# Designed Cell-Penetrating Peptide Constructs for Inhibition of Pathogenic Protein Self-Assembly

**DOI:** 10.3390/pharmaceutics16111443

**Published:** 2024-11-12

**Authors:** Mona Kalmouni, Yujeong Oh, Wael Alata, Mazin Magzoub

**Affiliations:** Biology Program, Division of Science, New York University Abu Dhabi, Saadiyat Island Campus, Abu Dhabi P.O. Box 129188, United Arab Emirates; yo612@nyu.edu (Y.O.);

**Keywords:** Alzheimer’s disease, alpha-synuclein, amyloid-beta peptide, cancer, mutant p53, neurodegeneration, Parkinson’s disease, prion, protein aggregation, tau

## Abstract

Peptides possess a number of pharmacologically desirable properties, including greater chemical diversity than other biomolecule classes and the ability to selectively bind to specific targets with high potency, as well as biocompatibility, biodegradability, and ease and low cost of production. Consequently, there has been considerable interest in developing peptide-based therapeutics, including amyloid inhibitors. However, a major hindrance to the successful therapeutic application of peptides is their poor delivery to target tissues, cells or subcellular organelles. To overcome these issues, recent efforts have focused on engineering cell-penetrating peptide (CPP) antagonists of amyloidogenesis, which combine the attractive intrinsic properties of peptides with potent therapeutic effects (i.e., inhibition of amyloid formation and the associated cytotoxicity) and highly efficient delivery (to target tissue, cells, and organelles). This review highlights some promising CPP constructs designed to target amyloid aggregation associated with a diverse range of disorders, including Alzheimer’s disease, transmissible spongiform encephalopathies (or prion diseases), Parkinson’s disease, and cancer.

## 1. Introduction

A range of degenerative conditions are associated with the misfolding and self-assembly of functional proteins and peptides into well-ordered aggregates [1]. These protein aggregates, known as amyloids, accumulate within tissues and various organs resulting in necrosis, subversion of the normal tissue architecture, organ dysfunction, and potentially death [2,3]. These so-called amyloid diseases include Alzheimer’s disease (AD), Huntington’s disease (HD), Parkinson’s disease (PD), transmissible spongiform encephalopathies (or prion diseases), and type II diabetes (T2D), which are mediated by the aggregation of the amyloid-β peptide (Aβ) and tau protein, huntingtin (HTT), α-synuclein (α-syn), prion protein (PrP), and islet amyloid polypeptide (IAPP, also known as amylin), respectively [1]. Amyloid aggregates share some common physical, tinctorial and structural properties; they are characterized by a predominantly β-strand secondary structure [4,5], high thermodynamic stability, insolubility in common solvents, and resistance to proteolytic digestion, a feature that impairs their effective clearance [2]. The presence of inflammation at sites of amyloid deposition is another pathological characteristic of these diseases, as tissue damage caused by amyloid fibrils elicits an immune response, producing proinflammatory cytokines [6,7]. Furthermore, recent evidence has shown that exposing cells to toxic amyloids can sometimes trigger a cascade of physiological changes; by engaging in aberrant interactions with cellular components, certain cellular events can be disrupted (e.g., impairment of the ubiquitin–proteasome system or disruption of membranes of intracellular organelles and their functions), ultimately leading to cellular death [1,4,8,9].

Important insights into amyloid fibrils at a molecular level, and how fibril formation relates to disease, have been provided by structural biology techniques, including nuclear magnetic resonance (NMR), transmission electron microscopy (TEM), X-ray crystallography, and cryo-electron microscopy (cryo-EM) [8,10]. Amyloid fibrils are unbranched and usually consist of two or more protofilaments that are often twisted around each other to form supercoiled rope-like structures [8,10]. Within each individual protofilament, the protein or peptide molecules are arranged in a manner to allow the polypeptide chain to form β-pleated sheet structures, which are oriented perpendicular to the long axis of the fibril [8,10,11]. Interestingly, since a protein can adopt a multitude of different conformational states within a living system, protein aggregation linked to disease has been found to originate both from globular proteins that fold to a compact tertiary structure in their non-aggregated state, including β2-microglobulin, transthyretin (TTR) and mutants of human lysozyme, and from intrinsically disordered proteins (IDPs) that are largely unstructured in solution, including α-synuclein, tau, Aβ and IAPP [5].

One of the most extensively studied amyloid diseases is AD, which is the most prevalent progressive neurodegenerative disorder and accounts for 60–70% of all cases of dementia [12]. AD is characterized by the deposition of extracellular Aβ amyloid aggregates, which constitute the main component of senile plaques found within the brain parenchyma of AD patients [13]. The pathogenic Aβ peptide is derived from the enzymatic processing of the transmembrane amyloid-β precursor protein (AβPP) by β- and γ-secretases through the amyloidogenic pathway; this entails β-secretase cleaving AβPP to generate a soluble AβPP_s_-β ectodomain and a membrane-associated carboxy-terminal fragment composed of 99 amino acids (C99) [14]. Subsequent processing of C99 by γ-secretase produces the APP intracellular domain (AICD) and Aβ [14]. Among the 39- to 43-residue peptide products of AβPP cleavage, Aβ_40_ and Aβ_42_ are the most pathophysiologically relevant isoforms; the 40-residue peptide (Aβ_40_) is the most abundant variant, whereas the 42-residue peptide product (Aβ_42_) is the most amyloidogenic and toxic isoform, as well as the principal Aβ species found in amyloid plaques [15,16,17,18,19].

Another pathological hallmark of AD is the intracellular accumulation of aggregated hyperphosphorylated tau as neurofibrillary tangles (NFTs) [20,21,22]. Under physiological conditions, soluble tau is involved in the assembly and stabilization of axonal microtubules, as well as intracellular trafficking [23]. These various activities are regulated by post-translational modifications of the tau protein, such as phosphorylation, oxidation, nitration, glycosylation, and ubiquitination [24]. In AD, on the other hand, tau is hyperphosphorylated within neurons, which induces a conformational change in the protein that exposes its microtubule-binding domain and results in its detachment protein from the microtubules [25]. This, in turn, increases the sensitivity of microtubules to severing proteins such as katanin, ultimately disrupting microtubule assembly [25]. Additionally, abnormal hyperphosphorylation of tau causes it to adopt a high β-sheet content, which initiates a nucleation-dependent aggregation process to form smaller soluble oligomers [26]. These oligomers further aggregate to form larger insoluble protofilaments, which associate in an antiparallel conformation, thereby developing into protofibrils [26]. Eventually, the protofibrils turn into paired helical filaments or straight filaments to form NFTs that are commonly observed during the later stages of AD [25,26].

Self-assembly of amyloid proteins is often studied using a combination of complementary methods, including measuring the fluorescence changes of amyloid sensitive dyes (e.g., thioflavin T (ThT) and Congo red), probing the secondary structure by circular dichroism spectroscopy (CD), Fourier transform infrared spectroscopy (FTIR) and NMR, measuring molecular tumbling by intrinsic fluorescence anisotropy (for amyloids containing fluorescent residues), and observing aggregate morphology using atomic force microscopy (AFM) and TEM [10,27,28]. These studies have revealed that amyloid fibril formation generally occurs via a nucleation-dependent polymerization mechanism, which consists of three distinct phases: a lag phase, an elongation phase, and a plateau or saturation phase [29,30]. The initial step of aggregation (nucleation phase) is thermodynamically unfavorable and involves completely or partially disordered monomers slowly self-assembling into minimum stable oligomers, which then act as seeds to further propagate exponentially via addition of monomers (elongation phase), until nearly all free monomers are converted into a fibrillar form (saturation phase) [8,31].

Although the details of amyloid formation-mediated cytotoxicity have not been completely elucidated, oligomeric forms, rather than the mature fibrils, are believed to play a key role in the pathogenesis of amyloid diseases [8]. A major contributor to the greater pathogenicity of oligomers is proposed to be the combination of a larger proportion of surface exposed hydrophobic residues and a higher diffusion coefficient, allowing them to diffuse more rapidly and form aberrant interactions with cellular membranes and other intracellular components more readily [5,32]. In support of the toxic oligomer hypothesis, prefibrillar soluble oligomers of Aβ were found to be substantially more cytotoxic, and to correlate far better with synaptic dysfunction and neurodegeneration, relative to either mature amyloid fibrils or monomers [33,34].

Despite the canonical cross-β structure being a shared characteristic, amyloids comprise a variety of structures, the molecular details of which are yet to be fully resolved [31,35]. This diversity is attributed to the precursors (to date, more than 40 disease-associated human proteins have been identified) having little similarity in amino acid sequence and native structure [5,8,36]. Consequently, the transition of these proteins to fibrillar species involves oligomeric intermediates that are highly heterogeneous in terms of size, stoichiometry, morphology, and toxicity [37]. Complicating matters is the fact that this heterogeneity is often observed with identical polypeptides, yielding distinct amyloid structures [35]. Thus, elucidating the molecular architecture of amyloids and the processes that generate them is critical for understanding the molecular basis of the associated diseases and identifying the pathogenic oligomeric form(s), which will facilitate the development of better diagnostics and more effective therapeutic interventions.

In this review, we discuss a contemporary innovation in the field of amyloid therapeutics, namely the development of cell-penetrating peptide (CPP)-based inhibitors of pathogenic protein self-assembly [38,39,40]. CPPs, also known as protein transduction domains (PTDs), are peptides (typically 5–40 amino acids) that possess the capacity to enter cells with high efficiency and low toxicity [41,42]. Importantly, a number of CPPs have also been shown to readily cross the blood–brain barrier (BBB) [43,44], intestinal walls [45], and intratumoral barriers [46]. Moreover, CPPs appear to retain these properties even when coupled to cargoes (e.g., proteins, nucleic acids, liposomes, and nanoparticles) many times their own molecular mass, which has resulted in the extensive use of these peptides as drug delivery vectors and therapeutic agents [41,42,47,48,49].

## 2. Therapeutic Strategies for Amyloid Diseases

The accumulation of protein aggregates that is characteristic of amyloidosis causes a wide range of symptoms [31,50]. As a result, many current treatments involve palliative care and drugs to manage and temporarily reduce these symptoms [50,51,52]. These treatments, however, have limited long-term therapeutic value since they focus mainly on the consequences of amyloid accumulation rather than addressing the underlying causes and inhibiting the progression of the disease. As an example, three of the FDA-approved drugs for AD (donepezil, rivastigmine, and galantamine) work by inhibiting cholinesterase from hydrolyzing acetylcholine, a neurotransmitter involved in memory and learning, while a fourth (memantine) is an N-methyl-d-aspartate (NMDA) receptor antagonist, which exerts its neuronal protective effects by inhibiting the excessive glutamatergic signaling thought to contribute to neuronal degeneration [51,53]. Interestingly, a combination therapy of memantine and donepezil was also approved as it has been reported to be more beneficial in managing moderate-to-severe AD symptoms, compared to monotherapy, by decreasing excitotoxicity induced by an overload of glutamate in the synapse and increasing the levels of acetylcholine at the synapse for a healthy synaptic transmission [54,55]. While yielding promising results in mouse models of AD, these drug candidates exhibited limited effectiveness in treating the cognitive symptoms in human subjects [53,56]. The marginal benefits provided by current therapies underline the urgent need to develop strategies that effectively stop or reverse the pathology of amyloid disorders.

One approach has been to develop molecules to inhibit production of the target amyloid protein or peptide [57]. For instance, in TTR amyloidogenesis, the homotetrameric TTR—that acts as a plasma transport protein for the thyroid hormone thyroxine and vitamin A—dissociates into dimers and then monomers, which subsequently misfold and self-assemble to form amyloid fibrils [58,59]. Since serum TTR is mainly synthesized in the liver, amyloidosis of the protein is typically treated by liver transplantation [60]. A potential alternative treatment strategy is to reduce the levels of TTR in the body using short interfering RNAs (siRNAs) [61]. However, these RNAi therapies are beset by major drawbacks that limit their successful application in a clinical setting, including low in vivo stability (due to susceptibility to endonucleases and exonucleases), potential off-target effects, and poor BBB and plasma membrane permeability [62].

In the case of AD, inhibitors have been developed against the two proteases, β- and γ-secretase, that generate the amyloidogenic Aβ peptides [63,64,65]. Although attenuating β- and γ-secretase activity has been shown to decrease the production of Aβ, only a few novel chemical compounds based on targeting the components of AβPP processing have progressed beyond the clinical trial stage [66]. One challenge is that β-secretase has a large catalytic pocket, necessitating the use of large molecules to block it; however, these molecules are unable to cross the BBB due to their large size [65,67,68]. Another major contributor to the lack of success of this strategy is that β- and γ-secretase regulate many other biological processes [69,70]. Of relevance, a major substrate of γ-secretase is the Notch receptor, and the Notch signaling pathway plays a key role in the differentiation and proliferation of many cell types [71,72,73,74]. Consequently, inactivation of these enzymes has been shown to cause a range of adverse effects, including accelerated cognitive decline and inhibition of T- and B-cell maturation, as well as hematological and gastrointestinal toxicity [75,76,77]. Finally, as is the case with TTR and other amyloid proteins, both Aβ and AβPP have important physiological functions—including in neuronal development (neuronal migration, neurite outgrowth, and synaptogenesis), synaptic function, BBB repair and the innate immune system (exhibiting antimicrobial activity against a range of bacteria, fungi, and oncogenic viruses)—and, thus, reducing the levels of the peptide, or interfering with the processing of its precursor protein, would result in detrimental side effects [78,79,80,81,82,83]. This was confirmed by reports that although homozygous AβPP-deficient mice were viable and fertile, they exhibited various abnormalities, including decreased body and brain weight, reduced locomotor activity and grip strength, spontaneous seizures, impaired behavioral performance and premature death [84,85]. Similarly, embryonic rodents injected with AβPP RNAi displayed signs of neuronal migration abnormalities [86].

A more viable therapeutic strategy is to target different steps in amyloid formation and clearance via stabilizers of the native conformation of the precursor proteins, inhibitors of fibrillogenesis, amyloid fibril disruptors, and promoters of amyloid clearance [51,65,87,88,89]. Specifically, in the case of TTR amyloidogenesis, small molecules were used to bind and stabilize the tetramer, thereby reducing its aggregation propensity [90,91,92]. Similarly, a variety of natural and synthetic molecules have been developed to bind to the aggregation-prone regions of Aβ and inhibit formation of its toxic oligomers [87,93,94]. An interesting subset of these molecules are designed α-helix mimetics, which are structured scaffolds that imitate the topography of the most commonly occurring protein secondary structure, as they present surface functionalities in a well-defined order to match the side-chain residues of a protein’s helical surface at positions i, i + 3/i + 4, and i + 7 [95,96]. The appeal of this approach is that the surface functionalities of α-helix mimetics can be conveniently manipulated to target specific protein–protein interactions (PPIs) at the interaction interface. α-helix mimetics have been applied to effectively constrain Aβ in a helical conformation, thereby potently inhibiting the amyloid peptide’s oligomerization and fiber formation [97,98,99].

In recent years, there has been increasing interest in developing peptides as amyloid inhibitors due to their desirable pharmacological properties: (i) ease and low cost of production; (ii) biocompatibility and biodegradability; (iii) greater chemical diversity than other biomolecule classes; and (iv) highly selective binding to specific targets, thereby minimizing off-target interactions and reducing the potential for toxicity [100]. These amyloid inhibitor peptides are either identified from library screens or are rationally designed (e.g., sequences derived from the target amyloid protein) [101,102]. However, a major drawback for peptide-based therapeutics is their poor tissue and cellular delivery efficiency [100]. This necessitates the use of delivery strategies to allow inhibitor peptides to overcome major physiological obstacles—such as the blood–brain and blood–cerebrospinal fluid barriers and the extracellular matrix of tissue, as well the plasma and intracellular membranes—and facilitate efficient delivery to target organs, cells, or subcellular organelles [100,103,104]. To this end, a variety of delivery approaches have been implemented, including electroporation, microinjection, and the use of nanocarriers [105]. However, electroporation and microinjection are cumbersome, time-consuming and can be damaging to tissues and cells [106], while a major concern with many nanocarriers is their inefficient release from endosomes following uptake by endocytosis (the primary cellular internalization route), which generally leads to degradative compartments, thus making this strategy ineffective for intact cytoplasmic delivery [107].

## 3. Cell-Penetrating Peptides (CPPs)

The history of CPPs dates back to the late 1980s when the *trans*-activator of transcription protein (TAT) of the human immunodeficiency virus type-1 (HIV-1) and the homeodomain of Antennapedia, a *Drosophila* transcription factor, were first recognized for their cell-penetrating propensities [108,109,110]. Extensive structure–activity relationship (SAR) studies to determine the minimum amino acid sequences required for efficient cellular internalization established the first CPPs: the TAT peptide and pAntp (or penetratin) [111,112,113]. Since then, a large number of CPPs have been identified or designed, which now include naturally occurring protein-derived peptides (e.g., TAT, pAntp, and pVEC (derived from vascular endothelial (VE)-cadherin, an endothelium-specific adhesion protein)), chimeras (fusions of sequences from two or more different sources, such as transportan (a combination of the N-terminal half of the neuropeptide galanin and the wasp venom-derived peptide mastoparan) and the tripartite Pep-1 peptide), and synthetic sequences (e.g., polyarginine, the model amphipathic peptide (MAP), and the cyclic cell-penetrating peptide 12 (CPP12)) [41,114,115].

CPPs are a class of diverse peptides that possess a broad range of physico-chemical properties [49,114,116]. The majority of CPPs carry a net positive charge at physiological pH due to the presence of basic amino acid residues, which facilitate the initial cellular binding of the peptides—via electrostatic interactions with negatively charged cell-surface constituents (such as heparan sulfate proteoglycans and phospholipid headgroups)—that is a prerequisite for cellular uptake [117]. Arginine-rich CPPs (e.g., TAT and oligoarginine) exhibit superior internalization efficiency compared to lysine-rich peptides, which is attributed to the capacity of the guanidinium side-group of arginine to form stable bidentate hydrogen bonds with anions, as opposed to the ammonium cation of lysine that forms only one hydrogen bond [114,118,119,120]. Interestingly, despite the established importance of basic amino acids for the cell-penetrating property, a few negatively-charged CPPs have also been identified or developed [121]. A notable example is the amphipathic proline-rich SAP(E), in which the arginine residues of the sweet arrow peptide (SAP) were replaced by glutamates [122]. Interestingly, SAP(E) binds preferentially to zwitterionic phospholipids, and the CPP’s internalization is mediated by structural transitions, from a polyproline II helix in solution, due to refolding or self-association at the membrane surface [123].

Despite extensive studies, the internalization mechanism of CPPs remains an area of debate. While it is generally accepted that the majority of CPPs are taken up by endocytosis, alternative endocytic uptake routes—from various forms of micropinocytosis (including clathrin-mediated and caveolae-dependent endocytosis) to macropinocytosis (a rapid and nonspecific form of fluid phase endocytosis)—have often been reported for different CPPs and, surprisingly, even for the same CPPs [117]. Complicating the picture further is the observation that some CPPs efficiently enter cells at 4 °C, or following depletion of the cellular ATP pool (using sodium azide/deoxyglucose), both of which inhibit all energy-dependent uptake processes [124,125]. This indicates an energy-independent internalization mechanism for these CPPs that involves ‘direct translocation’ across the plasma membrane [124,125,126]. Although the exact nature of the direct translocation mechanism remains unclear, various models have been proposed, including transient pore formation, inverted hexagonal lipid phase (or ‘inverted micelle’) formation, or membrane thinning [114,115,116]. These seemingly contradictory observations regarding the cellular uptake mechanism of CPPs are likely due, in large part, to the different parameters and conditions of the experiments performed, such as cell type, peptide concentration, characteristics (e.g., size and charge) of the cargo—which includes the fluorophore used as a probe in microscopy- and flow cytometry-based internalization assays—as well as the nature of the peptide–cargo coupling (e.g., covalent conjugation vs. complexation) [127,128,129]. Taken together, these disparate results indicate that the efficient cellular internalization of CPPs may be a consequence of their capacity to harness alternative mechanisms, the contributions of which to the overall uptake is dependent on the particular experimental conditions.

## 4. CPP-Based Amyloid Inhibitors

### 4.1. Prion Protein (PrP)-Derived CPPs

One of the earliest examples of CPP antagonists of amyloid aggregation are sequences derived from the prion protein (PrP) [130]. PrP is implicated in transmissible spongiform encephalopathies (TSE) or prion diseases, which are fatal neurodegenerative disorders that are characterized by spongiform degeneration of the brain, motor dysfunction, and dementia [131]. These include Creutzfeldt–Jakob disease (CJD) in humans, scrapie in sheep, and bovine spongiform encephalopathy (BSE) in cattle [132]. A hallmark of prion diseases, which can be infectious, sporadic or hereditary in origin, is the conversion of the endogenous cellular form of PrP (PrP^C^) into the misfolded, aggregated, and infectious scrapie isoform (PrP^Sc^) [131].

PrP^C^ is a glycoprotein expressed predominantly in neurons, neuroendocrine cells, and within the lymphoreticular system [133,134,135]. During processing, an N-terminal signal peptide (residues 1–22 in human and mouse PrP, and 1–24 in bovine PrP) is cleaved before the protein is trafficked to the cell surface, where it is attached via a C-terminal glycosylphosphatidylinositol (GPI) anchor to the extracellular side of the plasma membrane [136]. Notably, however, the N-terminal signal peptide is retained in some disease-associated forms of the protein [137,138,139,140]. Immediately following the signal peptide is a highly basic nuclear localization signal (NLS)-like hexapeptide sequence (KKRPKP). While NLS sequences are predominantly found in nuclear proteins, where they serve to direct these proteins to the nucleus, NLS-like sequences (typically hexapeptides with ≥4 cationic amino acids) are occasionally found in non-nuclear proteins, such as PrP [141]. The NLS-like sequence is not only important for the continuous endocytic recycling of PrP^C^, but also plays a role in the abnormal intracellular trafficking and localization of the protein in some prion disease-related cases, and deleting this hexapeptide prolongs survival of prion-infected mice [142,143,144]. Thus, similar to the N-terminal signal peptide, the NLS-like sequence is implicated in the pathogenesis of prion diseases.

We have previously shown that the N-terminal segments of unprocessed PrP, comprising the uncleaved N-terminal signal peptide and the cationic NLS-like sequence, from mouse and bovine PrP (denoted mPrP_1–28_ and bPrP_1–30_, respectively; Table 1), possess highly interesting and potentially disease-relevant properties. Both mPrP_1–28_ and bPrP_1–30_ function as CPPs that are readily internalized into cultured cells and can efficiently transport a whole host of sizeable cargoes across cellular membranes [138,145]. This suggests that these peptides may facilitate intracellular delivery of PrP^Sc^. Additionally, biophysical studies have revealed that mPrP_1–28_ and bPrP_1–30_ are not only aggregation prone, but can also destabilize membranes through transient pore formation [146,147,148]. Together, the properties of the PrP-derived CPPs provide a potential mechanism for the self-assembly, infectivity, and neurotoxicity associated with prion diseases.

Subsequent studies revealed that treatment with mPrP_1–28_ and bPrP_1–30_ significantly reduced PrP^Sc^ levels in prion-infected mouse neuronal hypothalamic cells, without affecting endogenous PrP^C^ levels [130]. Moreover, the peptides substantially delayed infection of healthy neuronal hypothalamic cells exposed to scrapie [130]. The inhibition of prion conversion was attributed to binding of the cationic NLS-like sequence of the PrP-derived CPPs to negatively charged residues of PrP^Sc^. However, the NLS-like sequence alone is poorly cell-permeable and requires the hydrophobic signal peptide to acquire its CPP activities and, in turn, anti-prion properties [165]. Structure–activity studies carried out in order to improve the CPP design yielded the following findings: truncating the signal peptide abolished the anti-prion effects; similarly, coupling the NLS-like sequence to conventional cationic or hydrophobic CPPs (e.g., pAntp, TAT or transportan) led to a loss of the anti-prion properties. On the other hand, replacing the uncleaved PrP N-terminal signal peptide with a shorter and more flexible signal peptide from the neural cell adhesion molecule-1 (NCAM1 residues 1–19), another plasma membrane-anchored glycoprotein, increased the anti-prion potency of the CPP [149]. Collectively, these results suggest a mechanism by which a signal peptide from a cell surface or secretory protein promotes the transport of the prion-binding NLS-like sequence to a subcellular location where it can inhibit prion conversion and propagation [149].

### 4.2. CPP Inhibitors of Aβ Aggregation

The PrP NLS-like sequence (PrP_NLS_) has been shown to selectively bind to a wide range of amyloid oligomers and fibrils via interactions with a common supramolecular feature of protein aggregates [166,167,168]. This has led us to hypothesize that PrP_NLS_-based CPPs could potentially inhibit the pathogenic self-assembly of other amyloid proteins and peptides. To test this hypothesis, we probed the effects of NCAM1_1–19_ coupled to PrP_NLS_ (CPP construct denoted NCAM1-PrP; Table 1) on Aβ amyloid aggregation and its downstream toxic effects [150,169]. We then further investigated the underlying concept of an amyloid-derived cationic segment fused with a hydrophobic signal sequence as a general design of amyloid inhibitors by replacing PrP_NLS_ with a corresponding positively charged sequence derived from the central hydrophobic region of Aβ (Aβ_16–20_: KLVFF) [150,170,171,172]. An additional lysine residue was added to the N-terminus of the Aβ_16–20_ (i.e., K-Aβ_16–20_: KKLVFF) in order to increase the number of cationic residues and improve solubility of the CPP construct (denoted NCAM1-Aβ; Table 1) [150,169].

Employing a number of complementary in vitro and in silico approaches, we extensively characterized the interactions of the two CPP constructs with Aβ [150]. Specifically, we probed the effects of the designed CPP constructs on the oligomerization, amyloid formation, and associated cytotoxicity of Aβ using established aggregation and cell toxicity/viability assays. Concomitantly, confocal fluorescence microscopy was used to monitor the cellular uptake and intracellular localization of Aβ in the absence and presence of the CPPs [150]. Finally, NMR and molecular dynamics (MD) simulations were utilized to determine the mechanistic and energetic details of the Aβ–CPP binding interactions. These studies demonstrated that the CPPs effectively target both extra- and intracellular Aβ and inhibit its oligomerization and amyloid formation by stabilizing it in a non-amyloid state, which serves to abrogate Aβ-induced neurotoxicity (Figure 1). Significantly, amyloid inhibition by NCAM1-PrP and NCAM1-Aβ occurred at a substoichiometric ratio (2:1 Aβ:CPP), underlining the anti-amyloid potency of the constructs [150].

Interestingly, similar results to those obtained with the NCAM1-PrP and NCAM1-Aβ CPPs were reported for the rationally designed D-enantiomeric peptide RD2, which contains a highly basic C-terminal segment (consisting of 5 × R) reminiscent of the PrP NLS-like sequence [151,173]. The use of D-enantiomers, in lieu of the naturally occurring and proteolysis-susceptible L-amino acids, confers a higher degree of proteolytic stability on the peptide and reduces its potential immunogenicity [174,175]. RD2 effectively inhibited Aβ amyloid aggregation and the associated downstream toxic effects in vitro, which is attributed to RD2 binding to Aβ monomers and stabilizing them in their native intrinsically disordered state [176,177]. Importantly, RD2 demonstrated promising anti-AD effects—including significant improvements in cognition and learning behavior, as well as a deceleration in neurodegeneration—in mouse models of the disease [176,178].

Another notable example of an Aβ-targeting chimeric CPP is the retro-inverso (i.e., D-retro-enantiomer) peptide inhibitor RI-OR2-TAT, which is composed of retro-inverso Aβ_16–20_ flanked by the solubilizing residues RG- and -GR (RI-OR2), and coupled to the retro-inverso TAT peptide [152]. RI-OR2 has been shown to effectively block Aβ oligomerization and fibrillation and rescue Aβ-associated cytotoxicity [152]. Moreover, the use of D-enantiomers with a reversed sequence and chirality ensures increased stability compared to the parent peptide, whilst retaining bioactivity [179]. However, RI-OR2 exhibited poor BBB permeability, necessitating coupling to the TAT peptide to facilitate localization to brain tissue [152,153]. Daily intraperitoneal injections of RI-OR2-TAT into APP/PS1 transgenic mice over 21 days led to a significant reduction in cerebral cortex Aβ oligomer levels, Aβ plaques count, activated microglial cells, and oxidative damage, with a concomitant increase in young neurons in the dentate gyrus [153]. These findings underline the potential of RI-OR2-TAT as a BBB-penetrating peptide inhibitor for AD that effectively decreases Aβ amyloid deposition in brain tissue and stimulates neurogenesis.

In a similar vein, a chimeric CPP construct was generated by combining the D-isomer forms of a 6-mer from the A2V variant of Aβ (Aβ1-6_A2V_) and the TAT peptide (a construct denoted as Aβ1-6_A2V_TAT(D)) [154]. The human Aβ_A2V_ variant protects heterozygous carriers from AD, which is likely due to the propensity of Aβ1-6_A2V_ to adopt a turn configuration at Glu3–Arg5 in its heterotypic interaction with wild-type (WT) Aβ, hindering the latter’s self-assembly and reducing the associated toxicity in neuronal cells [154,180]. The TAT sequence was added to improve cell and brain permeability of Aβ1-6_A2V_. Treatment of APPswe/PS1dE9 transgenic mice with Aβ1-6_A2V_TAT(D) for a duration of 2.5 months resulted in an increase in Aβ_42_ in the soluble fraction, and a concomitant decrease of the peptide in the insoluble fraction, along with a reduction in aggregated Aβ in both compartments, indicating disaggregation of amyloid deposits [154]. In addition, Aβ1-6_A2V_TAT(D) protected the mice against cognitive impairment [154]. Moreover, Aβ1-6_A2V_TAT(D) reversed synaptopathy induced by Aβ in vitro and in TgCRND8 mice, highlighting the CPP construct’s neuroprotective effects [181].

### 4.3. CPP Inhibitors of Tau Aggregation

The accumulation of NFT in the brain has been shown to strongly correlate with the loss of neurons and cognitive decline observed in AD, suggesting that the propagation of aggregated tau is a major driving force in the pathogenesis of the disease [182,183,184]. As a result, there is a growing body of research focused on tau as a viable and effective AD therapeutic target [185,186]. Some of these promising tau-targeted therapies include modulators of tau aggregation and inhibitors of tau kinases [187].

A recent example of a peptide inhibitor of tau aggregation is a CPP construct based on the aggregation-nucleating hexapeptide PHF6 (^306^VQIVYK^311^) in repeat 3 of the microtubule-binding domain of the protein [155,188]. Several modifications were made to optimize the inhibitory potential of the peptide, namely: (i) a hydrophobic residue, valine^309^, was replaced with lysine, to prevent self-association via electrostatic repulsion, and the substituted lysine was acetylated to increase its binding affinity to the target ^306^VQIVYK^311^ sequence via hydrogen bonding; (ii) proline^312^ was retained in its original position to maintain its role as a β-sheet breaker in order to further inhibit aggregation; (iii) polyarginine was added to the proline side of the peptide to facilitate its crossing of the BBB and subsequent internalization into target neurons; and finally (iv) the sequence was retro-inverted for increased stability and improved activity [155]. The peptide, named (RI)-AG03, effectively reduced the aggregation of recombinant tau by >90% in vitro, and significantly improved neurodegenerative and behavioral phenotypes in a *Drosophila* model of tauopathy [155]. Interestingly, coupling RI-AG03 to liposomes (composed of sphingomyelin, cholesterol, and DSPE-PEG-maleimide)—via click chemistry between a cysteine residue added at the C-terminus of the peptide and the maleimide groups of the liposomes—increased cellular association of the liposomes 3-fold, relative to peptide-free liposomes [189]. Following cellular uptake by macropinocytosis and an energy-independent mechanism (e.g., membrane fusion and subsequent direct translocation), RI-AG03 dissociates from its liposome carrier, which allows the peptide to escape entrapment in degradative organelles and increases its intracellular availability and, in turn, its therapeutic potential [189].

Other promising CPP inhibitors of tau self-assembly were designed to target a second aggregation-promoting hexapeptide motif, designated PHF6* (^275^VQIINK^280^), located in repeat 2 of the microtubule-binding domain [190,191]. For instance, a large peptide library was screened using mirror-image phage display, with D-PHF6* fibrils as a target, in order to identify D-enantiomeric peptides that inhibit the pathological aggregation of tau [156]. A D-enantiomeric peptide, MMD3, and its reversed-sequence analog, MMD3rev, bound to and inhibited fibrillation of both the PHF6* peptide and full-length tau by promoting formation of large off-pathway amorphous aggregates [156]. Moreover, MMD3 and MMD3rev were readily taken up by neuronal cells in vitro, and both peptides exhibited prolonged intracellular stability [156].

A rather unexpected addition to the group of PHF6*-targeting peptides is hepta-histidine (7H), which was originally identified as an inhibitor of the pathological interaction between mutant Huntingtin (mHTT), a key factor in the development of Huntington’s disease (HD), and Ku70, a critical neuronal DNA damage repair protein whose function is impaired by mHTT [157,158]. 7H binds preferentially to the PHF6* motif of tau, and potently inhibits aggregation of fragments of the protein in vitro [158]. Notably, a TAT-7H construct suppressed tau phosphorylation in induced pluripotent stem (iPS) cell-derived neurons carrying the disease-associated tau^P301S^ mutation [158]. Collectively, these studies underline the potential of CPP constructs as tau-targeted therapeutics.

### 4.4. CPP Inhibitors of α-Synuclein Aggregation

Parkinson’s disease (PD), the second most common progressive neurodegenerative disorder after AD and the most prevalent movement disorder, is characterized by resting tremor, bradykinesia, rigidity, and later-stage postural instability [192,193]. A hallmark of PD is the formation of cytoplasmic proteinaceous inclusions, Lewy bodies (LBs) and Lewy neurites (LNs), in dopaminergic neurons of the substantia nigra pars compacta, which ultimately leads to the loss of these neurons and the resulting striatal dopamine deficiency that is directly associated with the motor symptoms of the disease [194,195]. The major component of LB and LN inclusions is misfolded and aggregated α-synuclein (α-syn), suggesting that blocking, slowing down, or reversing aggregation of the protein is a potentially effective therapeutic strategy for PD [195,196].

Intriguingly, the paralog β-synuclein (β-syn)—which lacks a central aggregation-prone sequence within its non-Aβ component (NAC) domain that is present in α-syn—was found to inhibit α-syn self-assembly in a dose-dependent manner [197,198]. The inhibitory effects of β-syn were hypothesized to be due to interaction with a putative transient oligomeric intermediate of α-syn, stabilizing it and inhibiting it from developing into fibrils [199,200]. This prompted systematic mapping of the entire sequence of β-syn, which identified a short stretch of amino acids (β-syn_36–45_: GVLYVGSKTR) that specifically binds to α-syn [159]. A retro-inverso analog of β-syn_36–45_ (RI-β-syn_36–45_) was equally effective in inhibiting oligomerization and fibrillation of α-syn as the native peptide, but exhibited superior serum stability and cellular uptake in α-synuclein overexpressing cells [159]. Notably, administering the RI-β-syn_36–45_ in a *Drosophila* model of PD strongly ameliorated disease-associated behavioral defects in the treated flies and significantly reduced levels of aggregated α-syn in their brains [159].

Another promising class of amyloid inhibitors are cyclic peptides, which are typically composed of alternating d- and l-α-amino acids that adopt flat ring conformations and can stack, via intermolecular hydrogen bonding, to form cross β-sheet-like tubular structures [201]. A notable example is the cyclic d,l-α-hexapeptide, CP-2, which self-assembles into supramolecular structures that structurally and functionally resemble amyloids [160]. These similarities suggested that CP-2 may interact with and modulate the aggregation and associated toxicity of amyloid proteins/peptides. Indeed, CP-2 was found to interact strongly with Aβ and not only inhibit its amyloid aggregation, but also completely disassemble pre-formed Aβ fibrils, as well as rescue Aβ-induced cytotoxicity in vitro [160]. Extending the cyclic peptide approach towards PD, CP-2 was observed to predominantly bind to the N-terminal and NAC domains of α-syn that are essential for membrane binding and self-assembly, and alter the protein’s overall conformation [161]. Consequently, CP-2 not only decreased α-syn oligomerization and fibrillation, but also disassembled and remodeled pre-formed fibrils into off-pathway nontoxic amorphous co-assemblies [161]. Importantly, CP-2 exhibited cell-penetrating properties and reduced intracellular accumulation and cytotoxicity of α-syn in a neuronal cell line overexpressing the protein [161]. Although the in vivo efficacy of CP-2 is yet to be confirmed, these studies suggest that the cyclic peptide and its derivatives may serve as generic conformational inhibitors that cross-interact with a range of amyloidogenic proteins and peptides and inhibit their self-assembly into toxic oligomers.

Besides paralog-derived sequences and cyclic peptides, a fairly recent addition to the family of α-syn inhibitors are peptides originating from the small ubiquitin-like modifier protein, SUMO1, which modulates the interactions of a range of proteins, including α-syn [202,203]. A SUMO1-derived peptide, SUMO1 (1–55), bound and stabilized monomeric α-syn—mainly through two SUMO-interacting motifs (SIMs) located within the aggregation-regulating N-terminal region flanking the NAC domain—thereby suppressing aggregation of the protein [162,204]. Furthermore, SUMO1 (1–55) was readily taken up by cells in culture and effectively suppressed α-syn-mediated cytotoxicity in cell-based and *Drosophila* models of PD [162].

## 5. A Functional Bridge: Amyloids and Cancer

A rather unexpected addition to the family of pathogenic amyloid proteins is a subset of cancer-associated mutants of the tumor suppressor p53 protein [205,206,207]. Under cellular stresses, such as DNA damage or oxidative stress, p53 is activated to elicit the appropriate cellular response (e.g., DNA damage repair, cell cycle arrest or apoptosis) in order to suppress neoplastic transformation and inhibit tumor progression [208,209]. However, p53 is also the most mutated protein in cancer, with missense mutations in the protein occurring in >50% of all human cancers, and these mutations are associated with some of the most severe forms of the disease [209,210]. Consequently, p53 plays a crucial role in cancer research and is a key target for developing cancer treatments [210,211].

The p53 protein exists as a homotetramer under physiological conditions, with each monomer composed of discrete domains for DNA binding, tetramerization, and transcriptional activation [212]. p53 exhibits a high degree of structural flexibility that allows it to bind to several DNA sequences and interact with its numerous protein partners in its role as a sequence-specific transcriptional activator and master regulator of the cell [209,213]. A consequence of its conformational flexibility is that p53 normally exists in an equilibrium between native/folded, partially unfolded and aggregated states [164,214]. Crucially, ~90% of the cancer-associated p53 mutations occur within the thermodynamically unstable DNA-binding domain (DBD) [212,215], and many of these mutations decrease the domain’s stability further and prompt its unfolding, which leads to exposure of its normally hidden aggregation-nucleating subdomain (p53_251–258_: ILTIITLE) [164,216,217]. This, in turn, induces self-assembly of mutant p53 into amyloid-like aggregates within inactive cytosolic inclusions that often incorporate the WT isoform as well as its paralogs, p63 and p73 [164,214,217]. While p63 and p73 are rarely mutated and have partial functional overlap with p53, their incorporation in the cytosolic inclusions serves to suppress their regulatory functions, along with those of p53, culminating in uncontrolled proliferation, invasion, and metastasis [214].

Accumulating evidence implicates the amyloid-like aggregation of mutant p53 in not only the protein’s loss of tumor suppressor function, but also its oncogenic gain of function (i.e., acquisition of activities that promote tumorigenesis, metastasis, and chemoresistance), as well as the prion-like propagation of these phenotypes (in other words, the spread of mutant p53 aggregates to neighboring healthy cells to seed aggregation of endogenous WT p53) [164,206,207,209,214,218,219]. Thus, there is considerable interest in developing molecules that can effectively antagonize mutant p53 amyloid aggregation and rescue the protein’s tumor suppressor function, leading to apoptosis in mutant p53-bearing cancer cells, while exhibiting negligible toxicity to noncancerous cells [220,221]. Notable examples of such molecules include the cationic osmolyte acetylcholine chloride, the quinuclidinone compound PRIMA-1, the polyphenol resveratrol, arsenic trioxide (As_2_O_3_), and the bifunctional ligand L^I^, which simultaneously restores zinc binding in mutant p53 and inhibits its aggregation [222,223,224,225,226].

To develop a potential therapeutic strategy for mutant p53 aggregation, we adopted the protein mimetic-based approach that we had previously utilized to modulate various amyloid disease-associated PPIs [98,99,227,228]. We screened a library of oligopyridylamide-based protein mimetics—originally designed to antagonize amyloid formation associated with AD and T2D [98,99,227,228]—and identified a cationic tripyridylamide, denoted as ADH-6, which potently abrogated self-assembly of a p53 DBD-derived sequence, p53_248–273_, that contains both the aggregation-nucleating subdomain and the R248W mutation [229]. This mutation was chosen as it replaces the cationic arginine, an aggregation ‘gate-keeper’ residue [230], with the hydrophobic tryptophan, an aromatic residue with the highest amyloidogenic potential amongst all 20 proteinogenic amino acids [231]. Moreover, R248W is one of the most common p53 DBD mutations that occurs in a number of malignancies, including pancreatic cancer [215,232], which is associated with a very poor prognosis (global five-year survival rate is <5%) [233,234]. Subsequently, we showed that ADH-6 effectively targeted and dissociated cytosolic amyloid-like aggregates of a range of aggregation-prone p53 mutants in a variety of human cancer cells, which restored p53’s transcriptional activity, leading to cell cycle arrest and apoptosis (Figure 2) [229]. Notably, ADH-6 treatment substantially shrank xenografts harboring aggregation-prone p53 mutants but caused no toxicity to healthy tissue, leading to markedly prolonged survival [229].

A number of CPPs have also been used to target cancer-associated mutant p53 aggregation. For instance, polyarginine and its analog polyornithine significantly antagonized aggregation of a p53 DBD-derived sequence corresponding to the aggregation-nucleating subdomain and the aggregation-inducing R248Q mutation (p53_248–257_: QRPILTIITL) and inhibited the proliferation of aggregation-prone mutant p53-bearing cancer cells, which showed increased p21 expression that is indicative of reactivated p53, while having no adverse effect on the growth of WT p53 or p53-null cancer cells [163]. These effects can be attributed to the CPPs forming cation–π interactions with exposed aromatic residues of the partially unfolded mutant p53, which impedes the protein’s ability to self-assemble and shifts the folding equilibrium towards the active conformation [235].

The most notable mutant p53 aggregation inhibitor CPP construct is the ReACp53 peptide, which consists of a p53 DBD-derived sequence modified with the aggregation-suppressing I254R mutation and connected to the CPP R9 via a three-residue linker derived from the p53 sequence (RPI) [164]. Of relevance, the I254R mutation—which replaces a hydrophobic amino acid in the aggregation-prone core of p53 DBD with the ‘gate-keeper’ arginine residue—abrogates coaggregation of mutant p53 with the WT isoform and its paralogs, p63 and p73 [214]. ReACp53 was shown to block mutant p53 self-assembly by masking its aggregation-nucleating subdomain, which shifted the folding equilibrium towards the soluble state, leading to dissociation of the inactive amyloid-like cytosolic aggregates and accumulation of functional protein in the nucleus [164]. Treatment with ReACp53 restored p53 function in aggregation-prone mutant p53-bearing human ovarian and prostate cancer cells, which reduced cancer cell proliferation in vitro and halted tumor progression in vivo [164,236]. Interestingly, combining ReACp53 with the chemotherapeutic carboplatin resulted in synergistically enhanced efficacy against ovarian cancer cell lines in vitro and prolonged survival in a mouse model of high-grade serous ovarian cancer [237], indicating that ReACp53-mediated reactivation of the p53 pathway can potentiate DNA-damaging chemotherapeutics [220]. These studies suggest that targeting mutant p53 aggregation with amyloid inhibitor CPP constructs is a viable and effective cancer therapeutic strategy.

## 6. Conclusions

Accumulation of amyloid fibrils is a hallmark of a wide range of disorders, including neurodegenerative diseases such as Alzheimer’s, Parkinson’s and Huntington’s, as well as forms of mutant p53-associated cancers [1,207]. Currently, there are no treatments for effectively stopping or reversing the pathology of neurodegenerative amyloid diseases, which is a concern given that the burden of chronic neurodegenerative conditions is predicted to more than double over the next two decades [238]. In the case of cancer, mutant p53-associated forms of the disease are expected to result in the deaths of more than 500 million people alive today [239]. Consequently, there is a pressing need for new therapeutic interventions that can supplement or supplant current treatments for these cancers. CPPs appear to offer a viable and exciting option to help fill this pressing need owing to their attractive intrinsic drug-like properties: (i) ease and low cost of production; (ii) biocompatibility and biodegradability; (iii) greater chemical diversity than other biomolecule classes; (iv) highly selective binding to specific targets, which yields potent therapeutic effects while minimizing off-target interactions and reducing the potential for toxicity; and (v) capacity to overcome major physiological obstacles—such as the blood–brain and blood–cerebrospinal fluid barriers and tissue extracellular matrix, as well the plasma and intracellular membranes—to reach target organs, cells or subcellular organelles [100,101,102,103,104,240,241]. Indeed, a variety of amyloid inhibitor CPP constructs have been developed that have yielded potent therapeutic effects across a wide range of amyloid diseases, from inhibition of Aβ self-assembly and the associated cytotoxicity in neuronal cell lines, to dissociation of mutant p53 amyloid-like aggregates in inactive cytosolic inclusions in cancer cell lines, which restores the protein’s transcriptional activity, leading to cell cycle arrest and apoptosis [150,164]. Of note, these effects were also demonstrated in diseased tissue in in vivo models, typically with little-to-no visible toxicity to healthy tissue, culminating in markedly prolonged survival [150,153,154,164,178,181].

While the results obtained with the amyloid inhibitor CPP constructs are highly promising, as summarized in this review, some important questions remain to be addressed. For instance, prolonged treatment (>5 months) with the AD-targeted Aβ1-6_A2V_TAT(D) peptide increased the amyloid burden due to a shift in APP processing towards the amyloidogenic pathway, which is attributed to the D-isomer TAT sequence of the construct [154]. Thus, more extensive preclinical in vitro and in vivo testing is required to establish a comprehensive safety profile of these peptides, which is essential for the design of future clinical trials [242]. Moreover, optimizing the sequences of the amyloid inhibitor CPPs is critical for ensuring the translational success of these constructs. Of relevance, many of the amyloid inhibitor CPP sequences consist of hydrophobic and cationic segments (e.g., the Aβ inhibitors NCAM1–PrP, RD2, RI–OR2–TAT, and Aβ1-6_A2V_TAT(D); the tau inhibitor RI–AG03; and the mutant p53 inhibitor ReACp53), and this feature, which is necessary for the dual cell-penetrating and amyloid inhibition functions of these constructs, also facilitates their strong binding interactions with critical domains of the target proteins and peptides for potent antagonism of their self-assembly [150,153,154,164,235]. These results indicate a general underlying principle for the inhibition of pathogenic protein self-assembly, which can be leveraged to develop yet more effective CPP-based amyloid inhibitors.

Beyond their role as inhibitors of pathogenic protein self-assembly discussed here, a number of additional novel applications have been reported for amyloid-derived CPP constructs. For instance, two constructs consisting of two separate amyloid-forming hexapeptides—one derived from the microtubule-binding region of the human tau protein involved in formation of paired helical filaments of AD, the other from an amyloid-forming sequence from apolipoprotein A_1_—conjugated to polyarginine, exhibited selective cytotoxicity to cancer cells via formation of toxic oligomers by these cancer-targeting peptides (CTPs), relative to noncancerous cells [243]. More recently, CPP constructs with antimicrobial activity were generated by coupling amyloidogenic sequences of the bacterial ribosomal S1 protein to the TAT peptide via a flexible linker [244]. The amyloidogenic antimicrobial peptides (AAMPs) were shown to target and co-aggregate with the S1 protein, implicated in mRNA recognition and translation initiation and elongation [245], thereby inhibiting its physiological functions [244]. Collectively, these studies together with those summarized in this review, strongly suggest that the nascent field of amyloid-derived CPP constructs may yield a broad range of novel potential therapeutics, encompassing not only inhibitors of neurodegenerative diseases, but also anticancer agents and antimicrobial peptides.

## Figures and Tables

**Figure 1 pharmaceutics-16-01443-f001:**
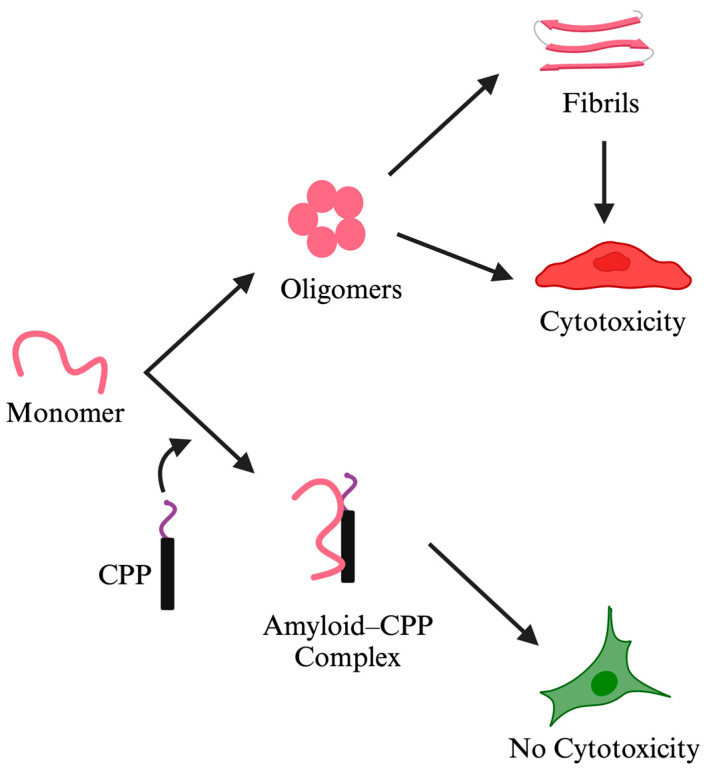
Designed cell-penetrating peptide (CPP) constructs inhibit amyloid aggregation and the associated cytotoxicity. Amyloid inhibitor CPP constructs are typically composed of distinct segments that contribute to the therapeutic effects (i.e., inhibition of oligomerization, fiber formation, and the associated cytotoxicity) and/or delivery properties (to target tissue, cells, and subcellular organelles). An example is NCAM1-PrP, which is composed of a hydrophobic signal peptide from the neural cell adhesion molecule-1 (NCAM1_1–19_: MLRTKDLIWTLFFLGTAVS) followed by a cationic nuclear localization signal (NLS)-like hexapeptide sequence from the prion protein (PrP_23–28_: KKRPKP) [149]. NCAM1-PrP was shown to effectively inhibit conversion of normal PrP^C^ into its disease-associated scrapie isoform of the protein (PrP^Sc^) [149], and to antagonize Aβ oligomerization, fiber formation, and the associated neurotoxicity [150]. The inhibition of pathogenic protein self-assembly is attributed to the NLS-like hexapeptide, but this sequence alone is poorly cell-permeable and requires the hydrophobic NCAM1 signal peptide to acquire its CPP properties [165]. Created in BioRender. Oh, Y. (accessed on 7 October 2024) BioRender.com/k09j761.

**Figure 2 pharmaceutics-16-01443-f002:**
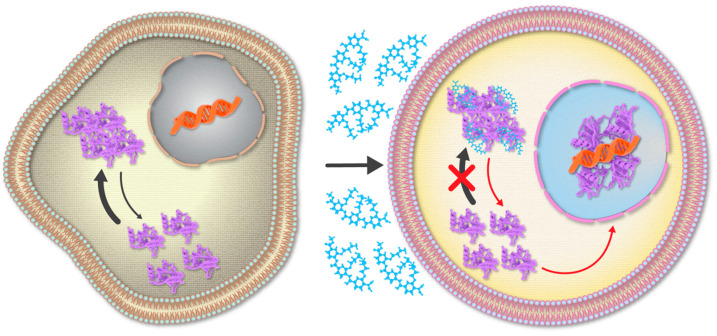
Amyloid inhibitors potently abrogate cancer-associated mutant p53 aggregation and restore tumor suppressor function. Under physiological conditions, p53 protein exists as a homotetramer, with each monomer composed of discrete domains for DNA binding, tetramerization, and transcriptional activation [212]. Approximately 90% of the cancer-associated p53 mutations occur within the thermodynamically unstable DNA-binding domain (DBD) [212,215], many of which decrease the domain’s stability further and prompt its unfolding and self-assembly into amyloid-like aggregates within inactive cytosolic inclusions [164,214,217]. Similar to the CPP construct ReACp53 [164], the protein mimetic ADH-6 (a cationic tripyridylamide) efficiently enters cancer cells, where it targets aggregation-prone p53 mutants and potently abrogates their self-assembly, which shifts the folding equilibrium towards the soluble state, leading to dissociation of the inactive cytosolic inclusions and accumulation of functional protein in the nucleus [229].

**Table 1 pharmaceutics-16-01443-t001:** A list of the amyloid inhibitor CPP constructs discussed in the article.

Peptide	Sequence	Target	Refs.
Mouse PrP_1–28_ (mPrP_1–28_)	MANLGYWLLALFVTMWTDVGLCKKRPKP	PrP	[138]
Bovine PrP_1–30_ (bPrP_1–30_)	MVKSKIGSWILVLFVAMWSDVGLCKKRPKP	PrP	[138]
NCAM1_1–19_–PrP_23–28_ (NCAM1–PrP)	MLRTKDLIWTLFFLGTAVSKKRPKP-NH_2_	PrP	[149]
NCAM1_1–19_–PrP_23–28_ (NCAM1–PrP)	MLRTKDLIWTLFFLGTAVSKKRPKP-NH_2_	Aβ	[150]
NCAM1_1–19_–K–Aβ_16–20_ (NCAM1–Aβ)	MLRTKDLIWTLFFLGTAVSKKLVFF-NH_2_	Aβ	[150]
RD2	ptlhthnrrrrr-NH_2_	Aβ	[151]
Retro-inverso–OR2 (RI–OR2)	Ac-rGffvlkGr-NH_2_	Aβ	[152]
Retro-inverso–OR2–TAT (RI–OR2–TAT)	Ac-rGffvlkGrrrrqrrkkrGy-NH_2_	Aβ	[153]
Aβ1–6_A2V_TAT(D)	dvefrhgggggrkkrrqrrr	Aβ	[154]
Retro-inverso–AG03 (RI–AG03)	Ac-rrrrrrrrGpkyk(ac)iqvGr-NH_2_	Tau	[155]
MMD3	dplkarhtsvwy	Tau	[156]
MMD3rev	ywvsthraklpd	Tau	[156]
7H	HHHHHHH	mHTT	[157]
7H	HHHHHHH	Tau	[158]
TAT–7H	YGRKKRRQRRRHHHHHHH	Tau	[158]
β-synuclein_36–45_	GVLYVGSKTR	α-syn	[159]
Retro-inverso–β-synuclein_36–45_ (RI–β-syn_36–45_)	rtksgvylvg	α-syn	[159]
CP-2	IJwHsK	Aβ	[160]
CP-2	IJwHsK	α-syn	[161]
SUMO1 (15–55)	DKKEGEYIKLKVIGQDSSEIHFKVKMTTHLKKLKESYCQRQ	α-syn	[162]
Polyarginine (and its analog polyornithine)	Rn	Mutant p53	[163]
ReACp53	(R9)RPILTRITLE	Mutant p53	[164]

Given are the CPP sequences, the target amyloid proteins/peptides, and the publications in which they first appeared.

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
