# Peer review of "Designed Cell-Penetrating Peptide Constructs for Inhibition of Pathogenic Protein Self-Assembly"

_pharmaceutics, 2024, doi:10.3390/pharmaceutics16111443_

Round 1

Reviewer 1 Report

Comments and Suggestions for Authors

This review article is clearly written. I have the following comments/suggestions:

(i) The purpose for developing CCP therapeutics is to treat diseases. The article however does not discuss the current status for such application. How far are we from a clinical trial? What are the challenges? These questions are probably discussed in general in other articles of the special issue but there are aspects specific to amyloid diseases.

(ii) A table of abbreviations would be helpful for readers.

(iii) Page 3, line13: “form aberrant interactions cellular membranes”?

Reviewer 2 Report

Comments and Suggestions for Authors

This review covers some promising CPP developments designed to target amyloid aggregation associated with a variety of disorders including Alzheimer's disease, transmissible spongiform encephalopathies (or prion diseases), Parkinson's disease, and cancer. But CPPs are also used to deliver antimicrobial peptides into cells to target aggregation, direct co-aggregation, as demonstrated in the discovery of novel amyloidogenic antibacterial peptides (see related reviews doi: 10.3390/ijms23105463). This should also be covered in the review.

The majority of the review is devoted to amyloid formation rather than to the analysis and development of CPPs. The introduction is entirely devoted to aggregation and amyloid formation. This is followed by another chapter on amyloid diseases. CPPs only appear on page 5. CPPs should be included in the introduction, since the title of the review is related to the development of CPPs: "Designed Cell-Penetrating Peptide Constructs for Inhibition of Pathogenic Protein Self-Assembly".

Reviewer 3 Report

Comments and Suggestions for Authors

The review titled "Designed Cell-Penetrating Peptide Constructs for Inhibition of Pathogenic Protein Self-Assembly" by Kalamouni et al. is generally well-written. While a significant portion of the work focuses on NCAM1 and PrP-peptides, these elements remain relevant to the overall objective of the review. It effectively highlights the main points in the introduction and organizes the peptide-based approaches according to their intended functions. Overall, the review is commendable in its current form.
